# 3D Vehicle Detection and Segmentation Based on EfficientNetB3 and CenterNet Residual Blocks

**DOI:** 10.3390/s22207990

**Published:** 2022-10-20

**Authors:** Alexey Kashevnik, Ammar Ali

**Affiliations:** 1St. Petersburg Federal Research Center of the Russian Academy of Sciences, SPC RAS, 199178 St. Petersburg, Russia; 2Information Technology and Programming Faculty, ITMO University, 197101 St. Petersburg, Russia

**Keywords:** autonomous driving, 3D object detection, localization, image processing, machine learning, vehicle classification, 3D segmentation

## Abstract

In this paper, we present a two stages solution to 3D vehicle detection and segmentation. The first stage depends on the combination of EfficientNetB3 architecture with multiparallel residual blocks (inspired by CenterNet architecture) for 3D localization and poses estimation for vehicles on the scene. The second stage takes the output of the first stage as input (cropped car images) to train EfficientNet B3 for the image recognition task. Using predefined 3D Models, we substitute each vehicle on the scene with its match using the rotation matrix and translation vector from the first stage to get the 3D detection bounding boxes and segmentation masks. We trained our models on an open-source dataset (ApolloCar3D). Our method outperforms all published solutions in terms of 6 degrees of freedom error (6 DoF err).

## 1. Introduction

3D vehicle detection and segmentation are some of the main parts of autonomous driving. For a self-driving vehicle, it is necessary to know the exact location and orientation of each other vehicles as dynamic objects on the scene. Based on this information, we can estimate the suitable speed, avoid crashes, and select the best path. It also reflects on the performance of other essential features such as motion prediction efficiency.

The 3D vehicle detection and segmentation necessity extend to other systems such as driver assistant and monitoring systems. Its usage is to alert the driver of dangerous situations and provides essential information about the surrounding environment.

Recent research on 3D Object Detection and 3D Segmentation could be classified according to the input type into two different branches. The first branch tackles LIDAR data (point clouds). Even though it has good results, it faces (running time, costs, and generalization) problems. On the other hand, monocular images as inputs (second branch) helped to design faster and low-cost systems trading accuracy. Some researchers proposed hybrid systems that depend on both (monocular images and LIDAR point clouds). Or generate point clouds from the monocular images using depth estimation or generative adversarial models.

Despite the fact that all these researchers tried to maximize the accuracy, they ignore the trade-off between the accuracy and running time. The main goal of these researches was aimed to maximize the mean average precision instead of minimizing the 6 DoF error.

Our proposed system uses a lightweight model (EfficientNetB3) with slight changes (CenterNet double convolution layers) to be able to achieve real-time performance on mobile devices. We aimed to minimize the 6 DoF error instead of maximizing the mean average precision because from our point of view it is better to find the rotation matrix and translation vector accurately than to maximize the accuracy of the bounding box.

In this paper, we tackle the problem of vehicle 3D detection and segmentation using localization in a 3D space from a 2D RGB monocular image and a database of 3D vehicle models. 3D accurate detection and segmentation for driving scene vehicles are one of the main challenges that stand against moving autonomous driving vehicles into production due to the lack of performance in terms of (accuracy and running time). On the other hand, integrating such systems into the driving assistant systems is a step forward to a fully automated driving recommendation system due to its role in understanding the surrounding environment as well as providing instructions for the automated parking process, estimating the relative speed, enhancing the depth estimation, and generating 3D models for moving objects (other vehicles).

Different approaches were published to solve the problem of 3D vehicle detection. Some of them depend on dividing the problem into subtasks. For example, 2D object detection/segmentation for the surrounding environment followed by 3D optimization. Other solutions build on one-stage approaches for 3D object detection with 3D convolution layers.

We proposed an approach that is based on end to end model for 3D localization. It is able to detect the 3D location of the vehicle center (x, y, z), its orientation (roll, pitch, yaw), and a second model that takes a crop around each recognized vehicle by the first stage as an input image to recognize the vehicle types in the scene (see Figure 1). We used the ApolloCar3D dataset for training that includes 75 different vehicle types. Using the output of our system (3D localization, orientation, and vehicle types), we visualized 3D models for the vehicles by replacing each of them with matched 3D models in the correct pose and location.

The main contribution of the paper is a modified EfficientnetB3 architecture (we added parallel double convolution blocks with skip connections “parallel residual blocks” inspired by the CenterNet architecture). The proposed architecture improved the state-of-the-art (GSNet) three degrees of freedom error for translation from 1.23 to 0.9 and three degrees of freedom error for rotation from 0.18 to 0.135. Our implementation will be published under an MIT license.

The structure of the paper is as follows. In Section 2, we present the related work on the topic of 3D vehicles, detection, segmentation, and localization. Section 3 shows the proposed method. Section 4 presents the evaluation and results. Section 5 describes the approach limitation. Section 6 describes the acknowledgments. The conclusion summarizes the paper.

## 2. Related Work

3D vehicle detection, segmentation, and localization are well-known problems in autonomous driving and driving assistant systems. A huge amount of research is directed to tackling these problems using different approaches.

Authors of the paper [1] proposed a monocular vision-based approach for 3D vehicle detection, localization, and tracking. The main key factor that helped the proposed approach to get competitive results even with LIDAR-based approaches is the multiframe optimization using the camera motion and tracking to improve the results.

Another approach published by [2] consists of three stages to enhance the monocular 3D localization: (1) a simple 2D object detection model to detect the vehicles; (2) a segmentation model to activate the pixels under the vehicle followed by; (3) a regression model that takes the segment results fused with the depth plane using a predefined ground plan parameters to find the 3D location. The research [3] proposed an end-to-end 3D object detection and trajectory prediction. The authors proposed the utilization of multiview representations of LiDAR that returns point clouds and camera images. They proposed a multifusion approach to maximize the benefits of all considered point clouds and images.

Another solution based on LIDAR and multiview representations is proposed by [4]. They published a new architecture called VPFNet that aligns and aggregates the image data and point cloud at virtual points that can bridge the resolution gap between the two sensors (LIDAR and multiview images).

In the paper, ref. [5] authors proposed an efficient two stages method for efficient 3D point cloud object detection. They parsed the point cloud data directly in the 3D space instead of using bird view projections by using a dynamic voxelization following the same processing pattern as pointwise methods. Another research study that relies only on LIDAR data cites SE-SSD is based on the self-ensembling of a single-stage object detector (SE-SSD). It contains a combination of teacher–student detectors where the teacher soft targets are filtered using an IoU-based matching strategy and formulate a consistency loss to align the student predictions with them.

Authors of the paper [6] proposed a novel single-stage 3D detection model (HVPR) that integrates voxel-based features and point-based features as well as an attentive multiscale feature module to extract scale-aware features considering the sparse and irregular patterns of a point cloud.

Back to monocular data images, the authors of the paper [7] proposed a pseudostereo approach with three virtual view generations (image-level, feature level, and feature clone). The authors proposed a disparitywise dynamic convolution with dynamic kernels to filter features adaptively from a single image for generating virtual image features and fixing depth errors. Authors of the paper [8] proposed a lightweight model for monocular data called progressive coordinate transforms, which is a localization boosting mechanism to refine the localization prediction. The main components of the proposed system are 2D object detection with depth estimation at the first stage, then the coordinates proposal followed by a 3D detector.

In research, ref. [9] authors proposed a method for incorporating the shape constraints into the 3D detection by learning a neural network to regress 3D correspondences of 2D keypoints, then apply geometric constraints to boost the detection performance.

Authors of the paper [10] proposed a new architecture suitable for temporal illumination that allows getting valuable information about the 2D object features that helps to optimize the 3D predictions through a frustum segment estimation. The main concept is divided into the following parts: (1) use a 2D object detector and (2) extract features from the cropped vehicles than using an attention mechanism for estimation of the 3D predictions. Authors of the paper [11] proposed to use the bird eye view to estimate the depth values and use the results to estimate the 3D detection from 2D detection, assuming that rotation will be only around one axis (yaw angle), making it weak against terrain scenes (mountains, hills, or even bridges).

Authors of the paper [12] proposed an extended version of the faster RCNN neural network that takes left and right images at once (from two cameras that take the driving scene). They use the detection of the sparse key points after the stereo region proposal network (RPN) to combine left and right boxes. They realign the 3D box using a region-based photometric (left and right ROIs).

Another approach proposed by the authors [13] is to use highly accurate 2D object detection. Using an energy minimization approach to place objects in the 3D space takes into account the idea that objects should be on the ground plane. Then, they are using the encoding of semantic segmentation, contextual information, size, and location priors as well as typical object shapes to project each candidate box to the image plane.

The authors of the paper [14] present a new approach for 3D object detection from point clouds by integrating both 3D voxel convolutional neural network (CNN) and point net-based set. The main key factor of the paper is replacing the convolutional pooling layers with ROI grid pooling to choose the features more efficiently.

Authors of the paper [15] proposed a graph neural network to detect objects from the LiDAR point cloud. They use the nearest neighbors as an encoding process to form the graph. An autoregistration approach is used to minimize the translation variance. The detection from multiple vertices is combined using a box merging and scoring operation.

Authors of the paper [16] published the current state of the art of end-to-end geometric and scene-aware networks. It constructs a 3D dense mesh representation to build a complex function for enhancing the 3D localization predictions.

In the paper ref. [17], the authors designed a key-points extractor for pose estimation for vehicles and humans. The paper is the current state of the art for car pose estimation on the ApolloCar3D dataset. They proposed to model for all key points that belong to an object as a graph using graph centrality measure to assign training weights to different parts of a pose.

Some researchers suggested complex methods for segmentation to improve its efficiency. In these papers [18,19,20], authors proposed to use an adaptive local prefitting energy function based on Jeffreys divergence to decrease the time complexity as well as increase the segmentation accuracy and solve the segmentation problems caused by the intensity inhomogeneity.

An Efficient Model for Autonomous Vehicles [21] is designed to target light devices. The proposed model uses a simple monocular camera and ultrasonic sensor to identify traffic signs to detect obstacles and avoid them. The proposed light system can run on a single Raspberry Pi.

In the paper [22], researchers proposed an adjustment to the 2D detectors such as SSD, YOLO, RetinaNet, etc. by adding another CNN decoder module so that the output of the proposed decoder will be the 3D parameters (dimension, orientation). They proposed a complex loss functions to train this multinet architecture.

Analysis of the current research paper shows that we should compare our results with the current state-of-the-art research [16] in terms of the six degrees of freedom error that is trained on the ApolloCar3D dataset.

## 3. Method

We proposed a method for 3D vehicle detection and segmentation. First, we built single-stage 3D vehicle localization and 2 stages 3D vehicle construction using a monocular image. We designed a neural network architecture to predict the position (*x*, *y*, *z*) and orientation (roll, pitch, yaw) for each vehicle in a monocular RGB image as well as another neural network architecture that can recognize types of vehicles. We used an ApolloCar3D dataset that includes 75 types of vehicles. Each car model dataset includes a 3D model which makes it possible to find the 3D box, 3D mask, and vehicle landmarks.

### 3.1. Dataset

For our training and test, we used an open-source dataset (ApolloCar3D) [23]. The data include RGB images for vehicle scenes with information about the location and orientation of each vehicle in every scene (see Figure 2). The type of each vehicle with a 3D model for it also provided with image masks to remove far vehicles from consideration. Camera characteristics are shown in Table 1.

### 3.2. Neural Network Architecture

We proposed architecture that consists of EfficientNetB3 with noisy student initial weights in parallel with four double convolution blocks. We join features with two steps of upsampling and a convolution layer as a head (see Figure 3).

We propose to crop the RGB input image (to keep only the bottom part with 50% height) to remove the upper part that contains unusable objects (sky, top of buildings, trees, signs, etc). Then the image is forwarded to EfficientNetB3 for extracting the features in parallel with four double convolution blocks. The double convolution blocks have two outputs: (1) the first one is taken after the fourth block and merged with EfficientNetB3 features by an upsampling layer; (2) the second one is the output of the upsampling is merged with a shallower block of the double convolution (third block skip connection) by another upsampling layer. The idea of the upsampling is to get a balanced output between near vehicles and far ones. Then, the output is passed to a convolution head layer that will produce the localization info to all detected vehicles.

#### 3.2.1. Double Convolutional Block

The double convolution block consists of a convolution layer followed by a batch normalization to stabilize the block and then a Relu activation to eliminate the negative values:(1)RELU(X)=max(0,x),
followed by the same combination (see Figure 4).

#### 3.2.2. EfficientNetB3

EfficientNet is a well-known architecture that achieved the state-of-the-art solution for multiple machine learning problems [24]. The hypothesis behind this architecture is based on balancing the scaling up the process for width height and depth for neural networks. We propose to use the EfficientNetB3 version to balance the good performance and the running time. As shown in Figure 5 the architecture consists of a convolution layer with swish activation followed by 26 convolution blocks. MBConvlotion is the inverted residual block (convolution layer then depth wise convolution followed by a convolution layer and the starting, ending of the block are connected with a skip connection). Then, a global average pooling to minimize the latent space dimensions.

Using the skip connection to connect the 3 double convolution blocks as shown in Figure 3, we propose to connect a residual block to the EfficientNetB3 architecture which we call partial ResNet EfficientNet architecture. The idea beyond this merging is to connect shallow layers with deep layers which will improve the performance, especially using upsampling for these connections to make the architecture more dynamic to the vehicle’s depth.

### 3.3. Vehicle Model Classification

Identifying the vehicle type is one of the most important key factors to getting accurate 3D detection and segmentation in our approach. Therefore we trained another neural network that will take the vehicle image cropped from the output of the suggested model and detect its type. That will make the inference slow but more accurate. we propose to use EfficientNetB3 with a small image size of 224 × 224 and a large batch size. Figure 1 in introduction section shows the adjustment to the system to enhance the accuracy. The classification model has been trained on Tesla V100, the batch size is 32, and using ReduceLROnPlateau schedule with an initial learning rate of $5\;\times 10^{-3}$ for 20 epochs, patience 2, and a factor of 0.1.

### 3.4. Loss Functions

The proposed model predicts different variables for each input image. The first variable is a mask that represents the vehicle’s centers which leads us to a problem similar to segmentation masks therefore we need to use BCE with logits for correction. Other variables include information about the 3D location (x,y,z) and orientation (yaw,pitch,roll). So, it is a regression value therefore we will use mean absolute error for correction. The output of the classification model will be the vehicle type; therefore, we need to use cross-entropy loss for correction.

As shown in Figure 3, The output of the system consists of a 2D mask for the car centers, so the first part of our loss function is a basic loss (BCE with logits) given by:$$loss_{mask}=mask_{gt}*log\left(mask_{pred}\right)+\left(1-mask_{gt}\right)*log\left(mask_{1-pred}\right),$$
where $mask_{gt}$ is the ground truth mask, $mask_{pred}$ is the mask prediction. For location and orientation correction we basically used mean absolute error at first:$$loss_{loc}=\left|loc_{pred}-loc_{gt}\right|$$
$$loss_{orient}=\left|orient_{pred}-orient_{gt}\right|.$$

Then for loss optimization, we observed that for the vehicle environment the most dynamic values are: x,z, and rotation around y(pitch). While other values are important but not so dynamic from the rules.

The vehicles will be always on the road plane.The vehicles will not be rotated around *z* axis (Yaw) unless it was an accident.The vehicle’s rotation over *x* axis (roll) is always small unless the scene was on terrain (mountains, hills, etc.).We can use the 2D mask to optimize the location.

So for the pitch angle, we took two outputs (sin,cos) to give it a priority in the loss corrections and to take mask values into account.
$$loss_{loc}=sum(sum(|loc_{pred}-loc_{gt}\left|)*mask\right)/sum\left(mask\right).$$

For car type prediction, it is a simple classification problem with 75 classes, we used cross entropy loss:$$loss=-\sum_{c=1}^{M}y_{o,c}*log\left(p_{o,c}\right),$$
where *M* is the number of classes, *y* is one if *c* is the correct class for observation *o*, *p* is the predicted probability for class *c* to be observation *o*.

### 3.5. Training Information

As shown in Figure 6 we take the input images, crop them to remove the upper part (sky and irrelevant objects), and build 2D masks for the vehicles centers to use them for correction.

We use 1600 × 700 image size for training, with batch size = 1 due to the GPU limit, (Tesla p100 16 GB). Learning rate equals $1\;\times 10^{-3}$, and StepLR scheduler (multiply LR by 0.1 every 3 epochs). We used AdamW optimizer because of its robustness against weight variation and large weight values.

## 4. Evaluation an Results

Based on the related work analysis we decide to use six degrees of freedom error (6 DoF error) instead of mAP for evaluation since it is more reasonable to understand the model behavior in terms of translation error and rotation error. In case the 6 DoF error is tiny and we have a highly accurate classification model then the mAP will be high because we are only substituting the 3D model inside the image using the translation vector and rotation matrix.

### 4.1. Classification Results

The classification model has 75 classes according to the used 3D models’ dataset. We got an accuracy of 69% for this multiclass single-label problem. In Figure 7, we present an example of how the system works and some samples of the data used to train the classification model.

### 4.2. Evaluation Metric

For location prediction, we take the best match on the ground truth and calculate the 3D Euclidean distance between the prediction and the ground truth in terms of translation for the vehicle center. For angular prediction (rotation) we calculate the mean squared error between the prediction and the ground truth. In Figure 8 we present a visualized example of our evaluation. The green shape is the prediction and the red one is the ground truth. Our evaluation metric will calculate the translation and rotation error between the vehicle centers.

We also defined the missing percentage rate that shows the number of vehicles in the scene that our model is not able to detect ignoring the far away vehicles as shown in Figure 8. Far vehicles (highlighted in red) don’t have predictions because they are not considered in the evaluation process (as they are also ignored by the dataset provider).

We present a comparison of our approach with GSNet [16] the current state of the art in terms of 6 DoF metric (see Table 2).

Considering of the far vehicles to express the percentage of the undetected ones on the scene caused a high miss rate of 21.6% that is caused by the number of far vehicles on the data (which were ignored during training using the masks), the very close vehicles (to the left or the right), and some cases where the vehicles are partially covered by other objects but presented on the ground truth.

In Figure 9, it shows the regression loss for our trained model.

### 4.3. Comparison in Terms of Loss

We present a comparison with other architectures trained using the same pipeline to present the strength of our proposed architecture (see Table 3). We present a visualization of substituting each detection by its classified model in Figure 10. In the Figure 11 white vehicles present misclassified vehicles. In Figure 12 we present a visualization comparison between our architecture and other neural networks over one image from the testing data. We present a visualization of substituting each detection with a random 3D model (without classification). Figure 13 shows visualization of results got by the proposed model over the test images from Apollo Car 3D dataset.

## 5. Limitations of the Proposed Method

Even though we got good results in terms of 3D localization and pose estimation but the method is still tied to multiple constraints and have some weakness. Because of the fixed number of 3D models (about 75 in the used dataset) which is a small number in comparison with the number of vehicle types in reality. For a new car that is not on our database, the model will classify it as a similar car which will affect the results of 3D model construction. Even if the vehicle is on the dataset the classification accuracy is due to the high similarity between the vehicle types especially when it is located in a faraway scene so there will be a huge loss in the feature context. Moreover, the neural network has a misrate equal to 21% which means the model will miss 1 out of 5 vehicles on each image. We have avoided heavy models to keep the running time in a reasonable range on light devices like Jetson Nano but the classification stage running time is dynamic and depends on the number of vehicles in a particular scene. We show further illustration of the proposed method in Figure 13. It shows more examples of 3D detection and segmentation using our proposed model.

## 6. Conclusions

We present a new architecture to perform vehicle 3D detection and segmentation based on 3D localization (find location and orientation) and 3D models for vehicles using the ApolloCars3D dataset. Our approach outperforms the current state of the art in terms of the translation error (0.9) and rotation error (0.135). We have also developed a classification model to enhance vehicle model type recognition. The main key factor to enhance the results was adding parallel double convolution blocks to the EfficientNetB3 architecture and joining the feature maps using multiupsampling and skip connection mechanisms, which is based on CenterNet architecture. For future enhancement we are planning to add a simple encoder/decoder component to provide an adaptive model to the camera constraints that will help move a variety of images to another unified domain before passing to the neural network. This component will use an attention mechanism to ignore the unneeded objects like (sky, trees and etc.) instead of cropping and adapting to the illumination changes. Moreover, adding Center Track to the system will minimize the inference time cost using tracking.

## Figures and Tables

**Figure 1 sensors-22-07990-f001:**
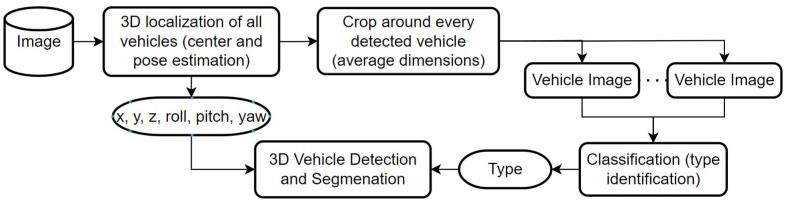
The proposed system methodology.

**Figure 2 sensors-22-07990-f002:**
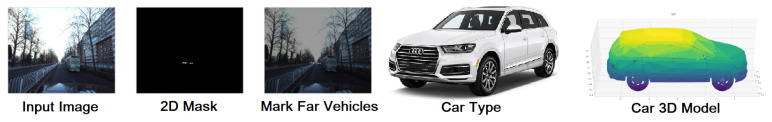
Training and testing data example.

**Figure 3 sensors-22-07990-f003:**
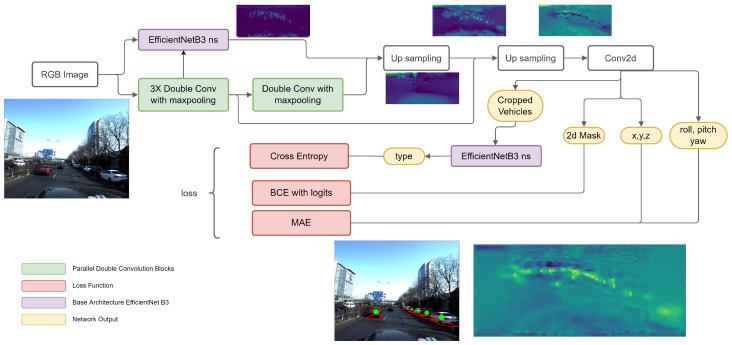
The proposed neural network model architecture.

**Figure 4 sensors-22-07990-f004:**
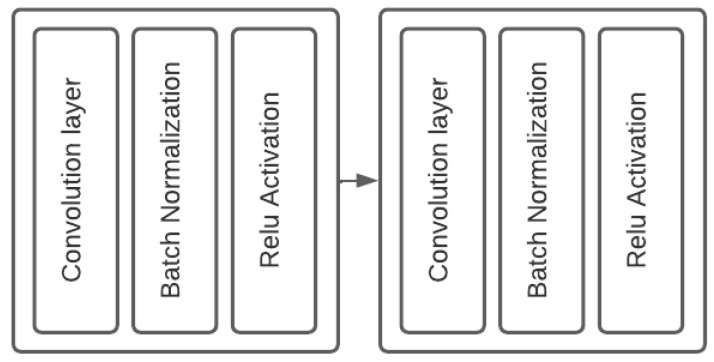
Double convolution Block, using kernel size 3 for all convolution layers, padding 1 and stride is 0 and the double convolution output in order (64, 128, 512, 1024).

**Figure 5 sensors-22-07990-f005:**
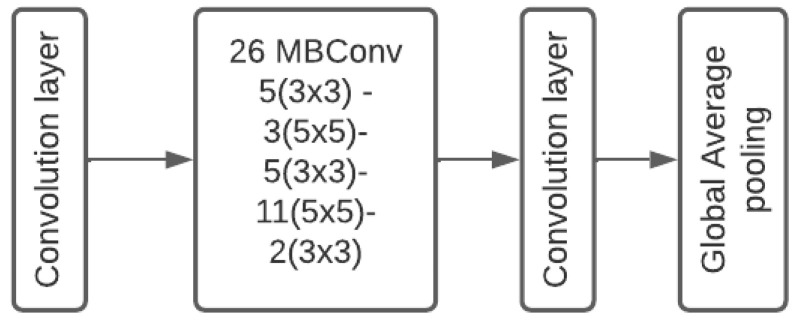
EfficientNetB3 architecture.

**Figure 6 sensors-22-07990-f006:**
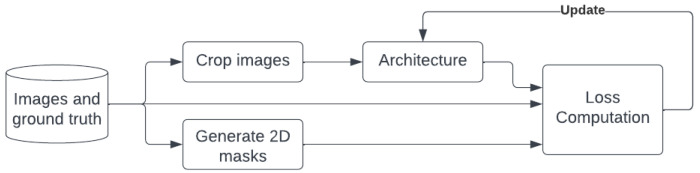
The proposed training pipeline.

**Figure 7 sensors-22-07990-f007:**
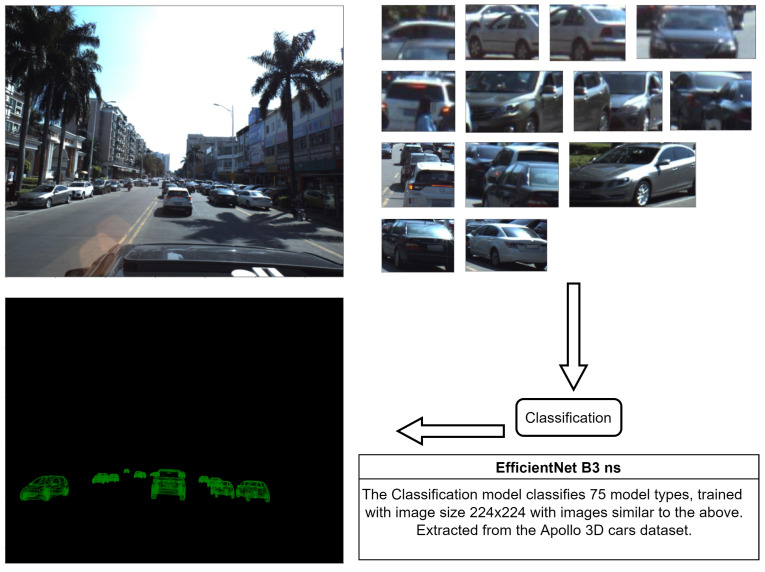
Visualization example for the classification process and data samples.

**Figure 8 sensors-22-07990-f008:**
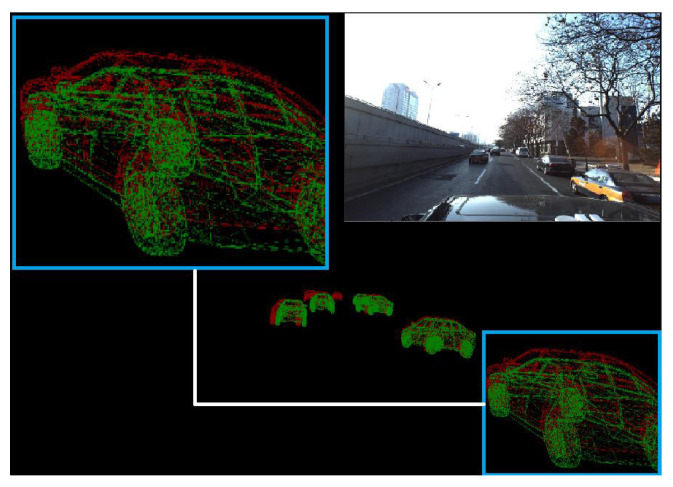
Evaluation metric computes the translation and rotation error between green predictions and red ground truth.

**Figure 9 sensors-22-07990-f009:**
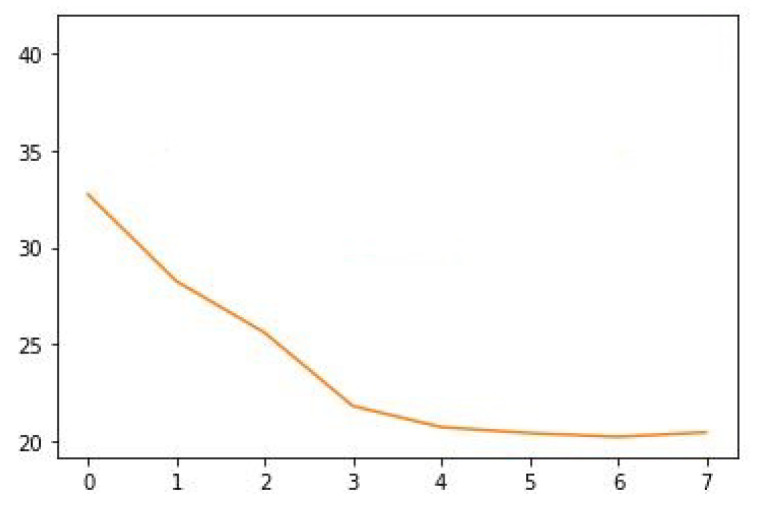
Regression Loss of our proposed model EfficientNet B3 with parallel convolutions (CenterNet).

**Figure 10 sensors-22-07990-f010:**
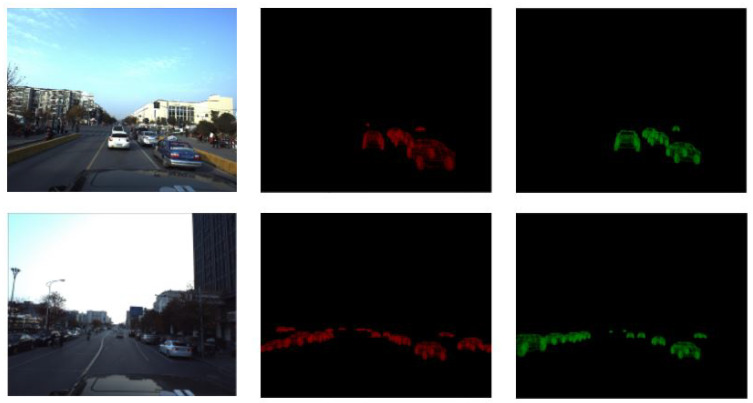
Visualization of model predictions (green) in comparison with the ground truth (red).

**Figure 11 sensors-22-07990-f011:**
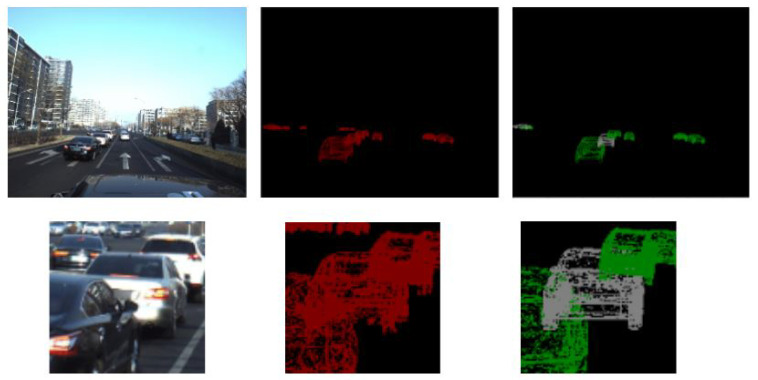
Visualization of model predictions (green) in comparison with the ground truth (red) where white predictions present the misclassified vehicles.

**Figure 12 sensors-22-07990-f012:**
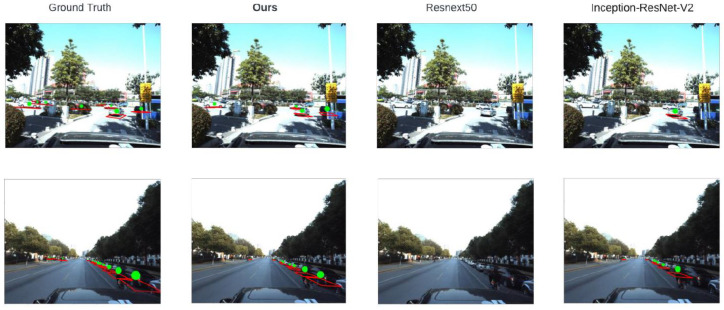
Results visualization in comparison with existing methods and ground truth.

**Figure 13 sensors-22-07990-f013:**
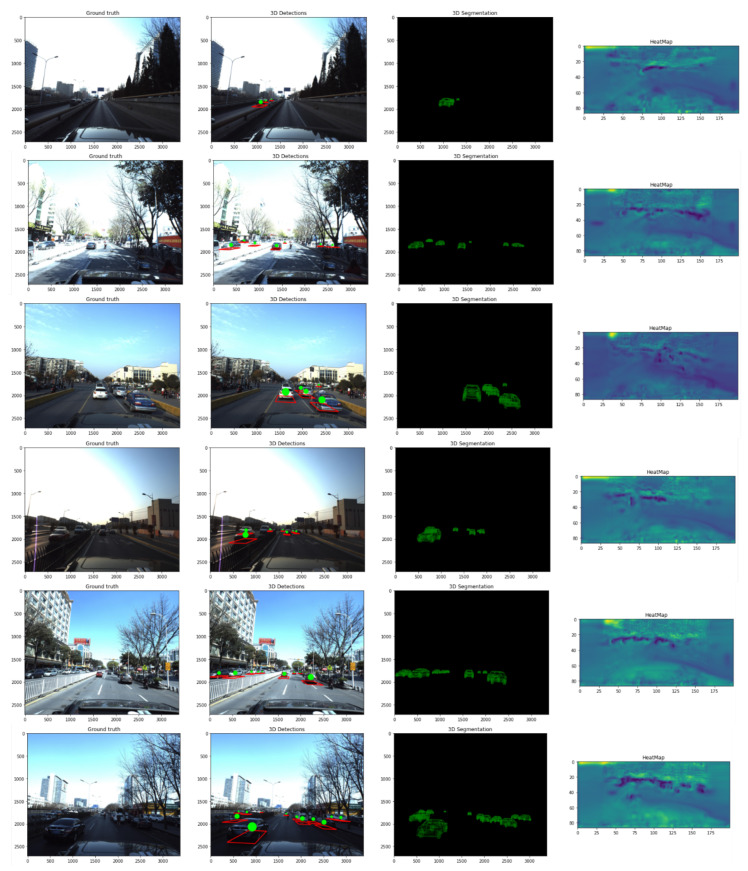
Visualization of results got by the proposed model over the test images from Apollo Car 3D dataset. From left to right (input image, 3D detection, 3D models substituted, and heatmap).

**Table 1 sensors-22-07990-t001:** Camera characteristics for ApolloCar3D dataset.

Focal_Length_x	2304.55
Focal_Length_y	2305.88
Optical_Center_x	1686.24
Optical_Center_y	1354.98

**Table 2 sensors-22-07990-t002:** 6 DoF error comparison.

Method	$3{DoF}_{err}\;T$	$3{DoF}_{err}\;R$	Miss Rate
GSNet	1.23	0.18	-
Ours	0.9	0.135	21.6%

**Table 3 sensors-22-07990-t003:** Regression loss comparison.

Model	Regression Loss
Inception-ResNetv2	0.38
ResNext-50	0.82
Center-resnext50	1.32
Ours	0.18

## Data Availability

Not applicable.

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
