# Peer review of "3D Vehicle Detection and Segmentation Based on EfficientNetB3 and CenterNet Residual Blocks"

_sensors, 2022, doi:10.3390/s22207990_

Round 1
Reviewer 1 Report
This paper proposes a 3D Vehicle Detection and Segmentation approach based on EfficientNetB3 and CenterNet residual blocks. In general, the topic is interesting and here are some comments:
1. The title of ‘car type’ in Figure 2 can be placed in the middle
2.The size of the title in Figure 2 is too small to be recognized easily
3. All titles under the figure can be better arranged if adjusted in the center of each line.
4. The size of pics in Figure 9, is too small for readers to know your result
5. The words in pics are also too small to recognized easily
6. More data should be compared with current method to show the advantage of your architecture, and the Visualization pics can be more convincing with existing data.
7. In Table 2, you can write more about the miss rate like the reason of such results.
8. You can make more comparison between different parameters and different architectures
9. In Conclusions and Future Work, you can talk more about your plan on improvement of your current model based on the data in experiments.
10. Other segmentation methods are suggested to added in the introduction section, such as active contour model, e.g., An active contour model driven by adaptive local pre-fitting energy function based on Jeffreys divergence for image segmentation, Expert Systems with Applications; A level set method based on additive bias correction for image segmentation, Expert Systems with Applications; A hybrid active contour model based on pre-fitting energy and adaptive functions for fast image segmentation, Pattern Recognition Letters
Author Response
We are thankful to the reviewers for the useful questions and valuable comments. Please, find in the attachment answers to the comments.

Reviewer 2 Report
The article used 3D localization and classification to perform 3D vehicle detection and its segmentation using single monocular RGB image. Two staged method is presented such as vehicle center detection and pose estimation and vehicle classification from driving scene. 3D location is estimated using EfficientNetB3 while the second stage classify the vehicle type and match it with predefined model.
The presented topic is very interesting and attractive, however, I have some major concerns that will further enhance the readability and structure of the paper.
The abstract should be modified as the current description doesn’t cover the main theme of the presented topic and the sentences lack sequential connections.
The main background and motivation is missing in the introduction section. Next, authors should explain the main challenges and problems in existing 3D vehicle detection and their segmentation method and highlight the solutions in light of their contributions in bullets. Next, the related work is very weak and need deep investigation to include the most recent literature and add the concluding remarks showing the limitations and complexities within them.
I went through proposed method where EfficientNetB3 is used without any major modification in the architecture or layers adjustment. Therefore, I do not find any novelty that could be claimed. Authors should highlight the main changes and the novelty.
Why authors used EfficientNetB3?
Most recent literatures such as 10.1109/TITS.2020.2980855, https://doi.org/10.1145/3561971
The figures are not well-presented and are dragged where the information cannot be easily seen.
Furthermore, author should assess the method performance with the existing state-of-the-arts.
Regarding the experimental section, the current visual and qualitative results are not enough and need more substances to be tested.
Author Response

(The authors gave the same response as above.)

Reviewer 3 Report
(1)I suggest that the author make adjustments to the writing of some sentences in the article, such as 2D object detection for the surrounding environment or 2D segmentation and (2) 3D optimization.
(2)The layout of Figure 3 is poor and I suggest the authors make adjustments.
(3)The author mentioned that the image size is first cropped, and the bottom 50% part is retained. Is this step suitable for removing the unusable object part (sky, building tops, trees, signs, etc.) in most images?
(4)In 6 DoF error comparison, I suggest that the author continue to compare the method proposed in this paper with at least three other methods, reflecting the advantages of the proposed algorithm.
(5)Can the author provide the loss curve corresponding to Table 3?
(6)I suggest that the authors collect data and build a database to make up for the lack of vehicle type, thus providing the algorithm's recognition efficiency.
Author Response

(The authors gave the same response as above.)

Round 2
Reviewer 1 Report
All my comments have been addressed.
Author Response
Thank you for pisitive feedback.
Reviewer 2 Report
This paper can be accepted.
Author Response
Thank you for pisitive feedback.
Reviewer 3 Report
1. Figure 7 looks blurry and I suggest replacing it with the original figure.
2. Can the authors give the regression loss curves of other models in Table 3 to show the advantages of the improved algorithm by comparison.
Author Response
We have replaced Figure 7 since it was blurry. Thank you for your comment.
Unfortunately, we don't have the regression loss curves of other models. We have the curve only for the our model. So, for other models we should reinitialized these experiments. Editor gives us only 3 days for minor revision response, unfortunatelly, it is not enought physically.
In any case, table 3 show the advantage of our in terms of regression loss.